# Dry Eye Treatment with Intense Pulsed Light for Improving Visual Outcomes After Cataract Surgery with Diffractive Trifocal Intraocular Lens Implantation

**DOI:** 10.3390/jcm13226973

**Published:** 2024-11-19

**Authors:** Takeshi Teshigawara, Miki Akaishi, Yuki Mizuki, Masaki Takeuchi, Kazuro Yabuki, Seiichiro Hata, Akira Meguro, Nobuhisa Mizuki

**Affiliations:** 1Department of Ophthalmology, Yokosuka Chuoh Eye Clinic, Yokosuka 238-0008, Kanagawa, Japan; 2Department of Ophthalmology, Tsurumi Chuoh Eye Clinic, Tsurumi, Yokohama 230-0051, Kanagawa, Japan; 3Department of Ophthalmology, Yokohama City University School of Medicine, Yokohama 236-0004, Kanagawa, Japan; akaishi.mik.pz@yokohama-cu.ac.jp (M.A.); yujimizumizu0302@gmail.com (Y.M.); takeuchi@yokohama-cu.ac.jp (M.T.); akmeguro@yokohama-cu.ac.jp (A.M.); mizunobu@yokohama-cu.ac.jp (N.M.); 4Department of Ophthalmology, Saiseikai Yokohamashi Nanbu Hospital, Yokohama 234-0054, Kanagawa, Japan; yabukazu@gmail.com; 5Department of Ophthalmology, Yokohama Sky Eye Clinic, Yokohama 220-0011, Kanagawa, Japan; s.and.e.hata@gmail.com

**Keywords:** cataract surgery, contrast sensitivity, diffractive trifocal intraocular lenses, dry eye, intense pulsed light, meibomian gland dysfunction

## Abstract

**Background/Objectives:** Meibomian gland dysfunction (MGD)-related dry eye aggravates postoperative visual outcomes in cataracts. Diffractive trifocal intraocular lenses (IOLs) decrease contrast sensitivity (CS). Intense pulsed light (IPL) improves tear film stability and ocular surface conditions in MGD-related dry eyes. We investigated the effect of preoperative MGD-related dry eye treatment combining manual meibomian gland expression (MGX) with IPL (IPL-MGX) on visual outcomes post-cataract surgery with diffractive trifocal IOL implantation. **Methods:** In this single-center, prospective, and open-label study, we enrolled 67 patients (134 eyes) with MGD-related dry eye undergoing cataract surgery on both eyes. Preoperatively, IPL-MGX was performed on one eye (IPL-MGX group) but not the contralateral eye (control group). Tear break-up time, high-order aberrations, and central superficial punctate keratopathy (C-SPK) were assessed. CS and corrected distance visual acuity were analyzed. Differences between groups were analyzed at 1 week, 1 month, and 3 months postoperatively. **Results:** The IPL-MGX group showed greater mean tear break-up time and lower mean high-order aberration and C-SPK values after preoperative IPL treatment and postoperatively (all *p* < 0.01). Postoperative CS was higher in the IPL-MGX group at 1 week (all spatial frequencies) (*p* < 0.01 [cpd = 2.9, 4.5, 7.1, and 10.2] and *p* < 0.05 [cpd = 1.1 and 1.8]); 1 month [2.9–10.2 cpd] (*p* < 0.01); and 3 months [4.5–10.2 cpd] (*p* < 0.01 [cpd = 10.2] and *p* < 0.05 [cpd = 4.5 and 7.1]) postoperatively. Mean corrected distance visual acuity was higher in the IPL-MGX group only postoperatively (*p* < 0.01). **Conclusions:** Preoperative MGD-related dry eye treatment using IPL-MGX enhances tear film stability, ocular surface conditions, and visual outcomes, potentially improving postoperative vision quality and patient satisfaction.

## 1. Introduction

Diffractive trifocal intraocular lenses (IOLs), which deliver incoming light simultaneously to three different foci, have become more popular owing to an increased desire for a spectacle-free life among patients [1]. However, this diffractive technology causes a loss of contrast sensitivity (CS), especially at higher spatial frequencies [2,3,4,5]. This inherent drawback can negatively affect the postoperative quality of vision [2,3,4,5]. Dry eye can also worsen CS [6,7,8]. The aggravation of CS in Meibomian gland dysfunction (MGD)-related dry eye is the worst at higher spatial frequencies [9]. In addition, visual acuity and reading speed can be particularly affected by CS at higher spatial frequencies [3,9]. Consequently, the combination of diffractive trifocal IOLs and MGD-related dry eyes can worsen the postoperative quality of vision and, as a result, decrease postoperative patient satisfaction [7,8,10].

MGD is defined as an abnormal meibum composition or secretion [11]. Lemp et al. reported that 86% of patients with dry eyes also had MGD, which contributed to the deterioration of tear film stability and ocular surface conditions [11]. The Prospective Health Assessment of Cataract Patient Ocular Surface study demonstrated that more than 60% of patients scheduled for cataract surgery had a short tear break-up time (TBUT) and that approximately 50% had positive staining in the central cornea [12]. Moreover, dry eye, including MGD, can be exacerbated by cataract surgery [13,14,15]. Thus, patients with MGD-related dry eyes are more susceptible to impaired quality of vision postoperatively, especially those with diffractive trifocal IOL implantation.

Intense pulsed light (IPL) is regularly used for dermatological treatments, such as rosacea and acne [16]. Toyos et al. first reported the efficacy of IPL treatment for MGD-related dry eye in 2015, and since then, multiple studies have further demonstrated the usefulness of IPL therapy for this condition [17,18,19]. Liu et al. found that in patients with MGD-related dry eyes, a therapy combining IPL and manual meibomian gland expression (IPL-MGX) was considerably more effective than MGX alone [20]. Leng et al. also reported that IPL alone was less effective than IPL-MGX combination therapy [21]. Considering these reports, it is speculated that preoperative MGD-related dry eye treatment with IPL-MGX may improve tear film stability, ocular surface conditions, and visual outcomes after cataract surgery with diffractive trifocal IOL implantation.

Previous studies have demonstrated the influence of preoperative dry eye treatment with different types of eye drops on postoperative tear film stability and ocular surface conditions in patients undergoing cataract surgery [22,23]. Teshigawara et al. reported that dry eye management with eye drops in the perioperative period was considerably effective in minimizing impaired quality of vision following cataract surgery with multifocal IOL implantation [8]. Furthermore, a combination of diffractive trifocal IOLs and MGD-related dry eyes can worsen the postoperative quality of vision [7,8,10]. Thus, they can decrease postoperative patient satisfaction [7,8,10]. Therefore, preoperative MGD-related dry eye treatment with IPL-MGX may enhance postoperative visual outcomes in diffractive trifocal IOL implantation cases, presenting an alternative to eye drops and potentially increasing patient satisfaction. However, previous studies have not examined the effectiveness of prophylactic preoperative MGD-related dry eye treatment with IPL-MGX for improving postoperative visual quality following cataract surgery with diffractive trifocal IOL implantation. Therefore, in this study, an investigation of the influence of preoperative MGD-related dry eye treatment with IPL-MGX on postoperative tear film stability, ocular surface conditions, and visual outcomes was conducted following cataract surgery with diffractive trifocal IOL implantation, following the hypothesis that IPL-MGX improves all outcome measures in this patient population.

## 2. Materials and Methods

### 2.1. Study Design and Patients

In this single-center, prospective, and open-label study, we enrolled Japanese nationals who underwent cataract surgery in both eyes followed by diffractive trifocal IOL implantation and who had been diagnosed with dry eye and MGD. The manuscript was prepared using the STROBE checklist for cross-sectional studies [24]. All enrolled patients were scheduled to undergo cataract surgery using diffractive trifocal IOLs and were selected based on the following inclusion and exclusion criteria on the same day they were diagnosed with cataracts affecting their vision. The inclusion criteria were as follows:

(1) All patients had both dry eye and hyposecretory MGD. Dry eyes were diagnosed based on the Japanese dry eye diagnostic criteria (TBUT ≤ 5 s and dry eye symptoms such as eye discomfort and visual disturbance) [25]. The dry eye symptoms were evaluated using the Japanese version of the Ocular Surface Disease Index [26]. Given that subjective symptom scores are not included in the Japanese dry eye diagnostic criteria, these symptoms were not quantified in the present study. Based on the Japanese MGD diagnostic criteria, hyposecretory MGD was diagnosed when the following three criteria were met: symptoms, abnormalities around the orifices, and findings indicating orifice blockage [27]. Abnormalities around the orifices were considered positive if at least one of the following three criteria was met: irregular lid margin, vascular engorgement, and mucocutaneous junction replacement, either anteriorly or posteriorly. Abnormalities indicating orifice blockage were judged as positive if both abnormalities indicating meibomian gland orifice obstruction (plugging, pouting, and ridging) and reduced meibomian secretion were observed [27].

(2) In relation to the preoperative screening results, patients were expected to experience postoperative vision improvement of at least 0.2 logMAR.

(3) Although there are no exact guidelines with respect to criteria for high-order aberrations (HOAs) for the usage of diffractive multifocal IOLs, the consensus of experts indicates that the root mean square of HOAs < 0.5 µm is recommended for diffractive multifocal IOLs [28]. Therefore, this was applied as an inclusion criterion in this study.

The exclusion criteria were as follows: (1) already receiving medication or using punctal plugs for dry eye before the start of study; (2) presence of blepharitis or active dermatological problems; (3) previously using contact lenses; (4) current usage of long-term medications; (5) having any of the following ocular histories: surgery, trauma, inflammation, scarring, dystrophy, or any condition causing ocular surface irregularity that could affect the quality of vision; (6) alteration of decision and opting for preoperative dry eye treatment, including eye drops, for the control eye, at any point during the preoperative period; (7) alteration of decision and opting for receiving postoperative dry eye treatment, including eye drops, for the control eye at any point during the postoperative period; and (8) presence of diabetes.

### 2.2. Treatment for MGD-Related Dry Eye

An ophthalmologist performed IPL-MGX. In all patients, IPL-MGX was performed only on one eye (IPL-MGX group) but not on the fellow eye (control group). All patients with dry eyes were randomly assigned to each group, using permuted block randomization, to receive IPL-MGX in either the right or left eye.

In the IPL-MGX group, IPL-MGX was administered in a series of four treatment sessions at 2-week intervals preoperatively. Patients were asked not to use new eye drops or any other type of medicine for dry eye or MGD during the IPL-MGX treatment period. In the control group, no dry eye treatment was implemented.

### 2.3. IPL-MGX Procedure

The IPL device (M22 IPL; Lumenis Be, Yokne’am Illit, Israel) was used in the triple-pulse sequence mode; the pulse duration was 6 ms, the pulse delay was 50 ms, the energy fluence range was 11 J/cm^2^ to 16 J/cm^2^, and the wavelength of the filter was 590 nm. The patients were advised to close both eyes, which were sealed with IPL-Aid eye shields (Honeywell Safety Products, Smithfield, RI, USA). An ultrasonic gel was applied to the targeted skin area, and the procedure was performed in two steps. In step 1, a double-pass technique of 12 impacts on the infraorbital/lower eyelid region with a 15 × 35 mm guide light was performed (Figure 1). In step 2, the single-pass technique of three impacts on the upper and lower eyelids with an 8 × 15 mm guide light was used (Figure 2). Approximately 5–10 min after the IPL procedure, MGX was performed on both the upper and lower eyelids using Yoshitomi Meibomian Gland Forceps (Charmant, Fukui, Japan). An oxybuprocaine hydrochloride ophthalmic solution (0.4%) was administered during this procedure to minimize patient discomfort.

### 2.4. Routine Pre- and Postoperative Eye Drop Administration

The following eye drops were administered to all eyes pre- and postoperatively: moxifloxacin eye drops (Vigamox) 4 times daily and 0.1% nepafenac ophthalmic suspension (Nevanac; both Alcon Laboratories, Inc., Fort Worth, TX, USA) 3 times daily for 3 days preoperatively. Postoperatively, moxifloxacin and 0.1% betamethasone sodium phosphate eye drops (Rinderon; Shionogi Pharmaceutical, Osaka, Japan) were administered four times daily for 2 weeks, as well as 0.09% bromfenac ophthalmic solution (Xibrom; ISTA Pharmaceuticals Inc., Irvine, CA, USA) twice daily for 1 month.

### 2.5. Surgical Technique

One surgeon performed all cataract surgeries. All patients underwent femtosecond laser-assisted cataract surgery (FLACS) (LenSx; Alcon Laboratories, Inc.) with implantation of a diffractive trifocal IOL (Clareon PanOptix; Alcon Laboratories, Inc.). The same parameter design was used for all patients undergoing FLACS. A 5.0 mm capsulotomy centered on the white-to-white diameter was made with 8.0 mJ of energy (spot and layer separation: 9 lm each). Nuclear fragmentation was performed using the chop and cylinder technique with 8.0 mJ of energy (spot and layer separation: 9 lm each). A 2.4 mm temporal incision of the clear cornea was manually made using a slit knife. Phacoemulsification and aspiration for cataract extraction were performed using the Centurion Vision System (Alcon Laboratories, Inc.). The diffractive trifocal IOL was carefully positioned at the center of the capsular bag. The interval between the first and second operations was approximately 1 week.

### 2.6. Examination of Pupil Size, Ocular Surface Conditions, and Visual Outcomes

TBUT was considered a measure of the tear film stability [29,30], superficial punctate keratopathy was in the center of the cornea (C-SPK), HOAs as measures of the ocular surface conditions [31,32,33,34], and corrected distance visual acuity (CDVA) and CS as measures of visual outcomes [6,35]. CDVA was measured by the same qualified optometrist using the Early Treatment Diabetic Retinopathy Study chart. Before IPL-MGX treatment was initiated, the photopic pupil size, TBUT, C-SPK, HOAs, and CDVA were examined for sample variations. After completion of four treatment sessions in approximately 1 week, TBUT, C-SPK, HOAs, and CDVA were analyzed to determine the effects of IPL-MGX on tear film stability, ocular surface conditions, and vision. In each group, TBUT, C-SPK, and HOAs were compared before cataract surgery, after IPL-MGX treatment, and 1 week after cataract surgery to determine the effects of cataract surgery on tear film stability and ocular surface conditions. At postoperative intervals of 1 week, 1 month, and 3 months, TBUT, C-SPK, HOAs, CDVA, and CS were analyzed to compare the effects of the treatment on tear film stability, ocular surface conditions, and visual outcomes between the IPL-MGX and control groups.

A fluorescent dye was used to assess TBUT and C-SPK. A fluorescent strip (Showa Yakuhin Kako Co., Tokyo, Japan) was moistened with saline and placed on the inferior bulbar conjunctiva. The mean time of TBUT was determined by timing it twice with a timer [36]. The degree of corneal staining at the center was assessed using the National Eye Institute/Industry Workshop method [37]. C-SPK was graded as follows: 0 = none, 1 = mild, 2 = moderate, and 3 = severe. The CASIA 2 (Tomey Corporation, Nagoya, Japan) measured both photopic pupil dilation and HOAs within a 4 mm area from the corneal center and provided a detailed representation of the cornea’s optical properties using Zernike polynomials. HOAs were assessed as a total value for the degree of third- to sixth-order aberrations, calculated as the root mean square [38]. The same experienced and certified optometrist examined CDVA with the Early Treatment Diabetic Retinopathy Study Chart. CS was measured under mesopic conditions (background luminance, 10 cd/m^2^) using the contrast glare tester, CGT-1000 (Takagi Seiko Co., Nagano, Japan), and analyzed monocularly based on six target ring sizes (Figure 3). The targets consisted of rings of differing sizes positioned 35 cm from the screen, and the assessment was conducted with near-corrected vision. The target visual angles were 6.3°, 4.0°, 2.5°, 1.6°, 1.0°, and 0.7°. The widths of the dark rings (2.9, 1.8, 1.2, 0.7, 0.5, and 0.3 mm) served as target details for calculating visual angles and cycles per degree (cpd). These widths corresponded to visual angles of 28.6°, 18.0°, 11.4°, 7.2°, 4.5°, and 2.9°, with corresponding cpd values of 1.1, 1.8, 2.9, 4.5, 7.1, and 10.4. Figure 4 presents the contrast threshold across 12 levels, varying from 0.01 to 0.45, for each visual angle of the targets. The log-CS was calculated using the contrast threshold values for statistical analyses [39].

### 2.7. Statistical Analysis

All statistical analyses were carried out using SPSS software (v. 29.0; IBM SPSS Statistics, Armonk, NY, USA). Statistical significance was set at *p* < 0.05, with Bonferroni correction applied for multiple comparisons. All variables were compared between the two study groups. The Shapiro–Wilk test was used to analyze the normality of variable distributions, revealing a non-normal distribution for all variables (detailed data are shown in Section 3). Further statistical tests were performed according to normality test outcomes. Pupil size, TBUT, C-SPK, HOAs, and CDVA at baseline before dry eye treatment were compared between groups using the Mann–Whitney U test [40]. TBUT, C-SPK, HOAs, and CDVA at baseline after dry eye treatment were compared between groups using the Mann–Whitney U test [40]. In each group, TBUT, C-SPK, and HOAs were compared before cataract surgery, after IPL-MGX treatment, and 1 week after cataract surgery using the Wilcoxon signed-rank sum test [41]. TBUT, C-SPK, HOAs, CDVA, and CS after cataract surgery were compared between the groups at 1 week, 1 month, and 3 months using the Mann–Whitney U test [40]. Post hoc analysis was performed using G*Power (v. 3.1.9.7) to analyze the sample size power. The post hoc power test indicated that the power of our analysis for the given sample size (n = 134) was 84.4% for a medium effect size (d = 0.5) with a significance level of 0.05.

## 3. Results

### 3.1. Baseline Patient Characteristics

This research included 134 eyes from 67 patients (32 men and 35 women) who were diagnosed with dry eye and MGD. The mean age was 76.6 ± 5.2 (range: 65–88) years. Study nurses double-checked that all participants followed the IPL-MGX treatment regimen, as well as the designated pre- and postoperative eye drop regimens. Deleterious effects related to IPL-MGX treatment, cataract surgery, or medications were not observed. The results of the Shapiro–Wilk test revealed a non-normal distribution for all variables. In the IPL-MGX group, the following values were observed: pupil size (*p* = 0.000 [baseline]); TBUT (*p* = 0.000 [baseline, after IPL-MGX, 1 week, 1 month, and 3 months]); C-SPK (*p* = 0.000 [baseline, after IPL-MGX, 1 week, 1 month, and 3 months]); HOAs (*p* = 0.009 [baseline], *p* = 0.007 [after IPL-MGX], *p* = 0.002 [1 week], *p* = 0.003 [1 month], and *p* = 0.007 [3 months]); CDVA (*p* = 0.000 [baseline, after IPL-MGX, 1 week, 1 month, and 3 months]); CS at 1 week (*p* = 0.000 [cpd = 1.1, 1.8, 4.5, and 10.2], *p* = 0.001 [cpd = 2.9], and *p* = 0.003 [cpd = 7.1]); CS at 1 month (*p* = 0.000 [cpd = 1.1, 1.8, 2.9, 4.5, and 10.2] and *p* = 0.002 [cpd = 7.1]); and CS at 3 months (*p* = 0.000 [cpd = 1.1,1.8, 4.5, 7.1, and 10.2]). Conversely, in the control group, the following values were observed: pupil size (*p* = 0.000 [baseline]); TBUT (*p* = 0.000 [baseline, after IPL-MGX, 1 week, 1 month, and 3 months]); C-SPK (*p* = 0.000 [baseline, after IPL-MGX, 1 week, 1 month, and 3 months]); HOAs (*p* = 0.006 [baseline], *p* = 0.004 [after IPL-MGX], *p* = 0.006 [1 week], *p* = 0.000 [1 month], and *p* = 0.003 [3 months]); CDVA (*p* = 0.000 [baseline, after IPL-MGX, 1 month, and 3 months] and *p* = 0.004 [1 week]); CS at 1 week (*p* = 0.003 [cpd = 4.5 and 10.2], *p* = 0.005 [cpd = 1.1 and 2.9], *p* = 0.007 [cpd = 7.1], and *p* = 0.008 [cpd = 1.8]); CS at 1 month (*p* = 0.001 [cpd = 4.5, 7.1, and 10.2] and *p* = 0.002 [cpd = 1.1, 1.8, and 2.9]); and CS at 3 months (*p* = 0.000 [cpd = 1.1,1.8, and 2.9] and *p* = 0.001 [cpd = 4.5, 7.1, and 10.2]). Table 1 presents the mean pupil size, TBUT, C-SPK, HOA, and CDVA values at baseline (before IPL-MGX treatment) for the IPL-MGX and control groups. The mean values at baseline were not significantly different between the two groups, suggesting the absence of sample variation between the groups.

### 3.2. Tear Film Stability, Ocular Surface Conditions, and CDVA After IPL-MGX Treatment but Before Cataract Surgery

Table 2 presents the mean TBUT, C-SPK, HOA, and CDVA values approximately 1 week after IPL-MGX but before cataract surgery in the two groups. We observed significant between-group differences in the mean values of tear film stability and ocular surface conditions (TBUT, C-SPK, and HOAs) (all *p* < 0.01) but not in the CDVA values (*p* = 1.00).

### 3.3. Within-Group Comparisons Between the Time Points After IPL-MGX Treatment but Before Cataract Surgery and 1 Week After Cataract Surgery

Table 3 shows the TBUT, C-SPK, and HOA values before (after IPL-MGX treatment) and 1 week after cataract surgery in the control (Table 3a) and IPL-MGX (Table 3b) groups. All variables significantly worsened in both groups (*p* < 0.01).

### 3.4. CDVA After Cataract Surgery

Figure 5 shows the CDVA values in the IPL-MGX and control groups at 1 week, 1 month, and 3 months postoperatively. The CDVA was significantly better in the IPL-MGX group than in the control group at each time point (*p* < 0.01 [1 week and 1 month] and *p* < 0.05 [3 months]).

### 3.5. Contrast Sensitivity After Cataract Surgery

Figure 6a–c illustrate the mean log-CS in the IPL-MGX and control groups at postoperative intervals of 1 week, 1 month, and 3 months, respectively. At 1 week, the differences between the two groups were significant at all spatial frequencies (*p* < 0.01 [cpd = 2.9, 4.5, 7.1, and 10.2] and *p* < 0.05 [cpd = 1.1 and 1.8]; Figure 6a). At 1 month, statistically significant differences between the groups were observed for cpd 2.9, 4.5, 7.1, and 10.2 (*p* < 0.01 [cpd = 4.5, 7.1, and 10.2] and *p* < 0.05 [cpd = 2.9]) but not for cpd 1.1 and 1.8 (Figure 6b). At 3 months, the differences between the groups were significant for only cpd 10.2 (*p* < 0.01; Figure 6c).

### 3.6. Tear Film Stability and Ocular Surface Conditions After Cataract Surgery

Table 4, Table 5 and Table 6 present the mean TBUT, C-SPK, and HOA values, respectively, in the IPL-MGX and control groups at the postoperative timepoints of 1 week, 1 month, and 3 months. The mean TBUT in the IPL-MGX group was significantly higher, and the mean HOA and C-SPK values in the IPL-MGX group were significantly lower than those in the control group at each time point (*p* < 0.05).

## 4. Discussion

Evidence of the effect of preoperative MGD-related dry eye treatment with IPL-MGX on postoperative measures of tear film stability (TBUT), ocular surface conditions (C-SPK and HOAs), and visual quality (CDVA and CS) in patients undergoing cataract surgery with diffractive trifocal IOL implantations is limited. Our findings indicate that preoperative treatment of MGD-related dry eye with IPL-MGX could alleviate the negative effects of cataract surgery on postoperative tear film stability, ocular surface conditions, and visual quality.

Previous research has shown the effectiveness of preoperative dry eye treatment with eye drops, such as cyclosporine 0.05%, diquafosol ophthalmic solution, and rebamipide, on postoperative tear film stability, ocular surface conditions, and quality of vision [8,23,42]. In addition to dry eye treatment using eye drops, modern physical treatments, such as LipiFlow thermal pulsation (Johnson & Johnson, Jacksonville, FL, USA), Tixel thermomechanical devices (Novoxel, Israel), and IPL, have also been used, especially for MGD-related dry eyes. Lane et al. [43], Toyos et al. [17], and Safir et al. [44] were the first to report the successful establishment of MGD treatment with LipiFlow thermal pulsation, IPL, and a Tixel thermomechanical device, respectively. Park et al. reported the efficacy of preoperative MGD-related dry eye treatment with LipiFlow thermal pulsation on postoperative tear film stability and ocular surface conditions [45]. Similarly, Ge et al. described the benefits of MGD-related dry eye treatment with IPL on postoperative tear film stability and ocular surface conditions [46]. However, these researchers only investigated postoperative tear film stability and ocular surface conditions, not postoperative visual outcomes [45,46]. Although Martínez-Hergueta et al. investigated the efficacy of MGD-related dry eye treatment with IPL on visual outcomes, they used laser refractive surgery and not cataract surgery [47]. The influence of the Tixel thermomechanical device on postoperative conditions has not been investigated. To the best of our knowledge, our study is the first to demonstrate the efficacy of preoperative MGD-related dry eye treatment combined with physical MGD treatment (IPL-MGX) for improving postoperative visual outcomes after cataract surgery with diffractive trifocal IOL implantation.

The hyposecretory and obstructive types of MGD are the most common. A feature of these meibomian gland abnormalities is terminal duct blockage [48]. Anatomical degeneration and pathophysiological changes can occur in MGD, and these conditions alter the lipid composition of the tear film, accelerating evaporative dry eye [49]. MGD-related dry eye reportedly worsens after cataract surgery [15,45]. However, the mechanisms underlying postoperative aggravation of MGD remain unclear. In the present study, variables indicative of tear film stability and ocular surface conditions considerably worsened after cataract surgery, regardless of the history of preoperative dry eye treatment with IPL-MGX. In a cohort of Japanese patients with planned cataract surgery, 74.5% were diagnosed with MGD [50]. MGD-related dry eye can aggravate CS, especially at higher spatial frequencies [9]. Additionally, diffractive multifocal IOLs can also reduce CS at higher spatial frequencies [3,4,5]. Therefore, CS at higher spatial frequencies may likely be markedly affected by a combination of MGD-related dry eye and implantation of diffractive trifocal IOLs. Additional concerns are that visual acuity and reading speed with small letters are related to CS at higher spatial frequencies [3,51]. These findings indicate that the management of MGD-related dry eye is critical in patients, especially with diffractive trifocal IOL implantation.

Current MGD treatment options comprise eye drops and physical therapy, the latter including lid hygiene, lid heat application, lid massage, MGX, LipiFlow thermal pulsation, and IPL therapy. As the IPL-MGX combination is reportedly more effective than either IPL or MGX alone [20,21], we used an IPL-MGX combination therapy in the present study. Despite encouraging results, the exact mechanisms underlying IPL therapy remain unclear. We speculate that the following mechanisms might be involved [52]. The energy of the IPL flash is absorbed by the target tissue, and the temperature increases in this area. An increase in temperature in the meibomian glands can liquefy lipids and improve glandular secretion and excretion. However, other substances that block the meibomian gland, such as keratin, cannot be liquefied by an increase in temperature, and keratin should be physically removed using MGX. Therefore, IPL and MGX can be regarded as complementary methods. Additionally, the absorption of light by red blood cells can cause the collapse of telangiectatic vessels of the lid margin. Blocking dilated capillaries and reducing the release of inflammatory mediators lessens the inflammatory response of acinar cells and reduces edema. Moreover, the bacterial load is reduced, and the macro- and microstructures of the meibomian glands can be improved [53].

In the present study, postoperative tear film stability (TBUT) and ocular surface conditions (C-SPK and HOAs) were markedly better in the IPL-MGX group than in the control group. These results correspond to those of previous studies that also used modern physical therapies such as IPL or LipiFlow thermal pulsation [45,46]. Previous studies have suggested that dry eye, including MGD-related dry eye, could reduce visual quality owing to tear film instability and corneal surface irregularity [54,55]. Several studies may provide reasons for this observation. Tear film instability can increase the magnitude of HOAs, which, in turn, can affect CS [56,57]. Patients with dry eyes and C-SPK have considerably higher HOA values than those without C-SPK [58], and C-SPK is responsible for decreased CS in patients with dry eyes [6,57]. The relationships among these variables indicate that improving tear film stability and ocular surface conditions in patients with dry eyes can alleviate the decrease in CS. In the current study, CS in the IPL-MGX group, which showed postoperatively better tear film stability and ocular surface conditions, was markedly better than that in the control group. This finding was considerably more pronounced at higher spatial frequencies. At lower spatial frequencies, the efficacy in the IPL-MGX group was considerably greater in the early postoperative period but slightly declined over time. This result is in line with previous findings, in which dry eye and its influences were more severe in the early postoperative stage [13,59].

However, at higher spatial frequencies, CS in the IPL-MGX group was markedly better than that in the control group throughout the study period. Additionally, the postoperative CDVA in the IPL-MGX group was considerably better than that in the control group throughout the study period. These results also support those of previous studies that reported a negative effect of dry eye on visual acuity and CS [35,60]. The postoperative CS and CDVA results in the present study can be explained by the following scientific evidence. CS at high spatial frequencies is responsible for the detection and identification of small objects, which corresponds to the measurement of visual acuity [4,51,61] and reading speed [3]. Previous studies have reported that the CS at high spatial frequencies is considerably lower in patients with diffractive multifocal IOLs than in patients with monofocal IOLs [4,62]. Therefore, the findings of this study indicate that preoperative IPL-MGX treatment can be a promising method to improve postoperative visual outcomes and increase postoperative patient satisfaction in cataract surgery with diffractive trifocal IOL implantation.

In this study, sustained improvement in the quality of vision over time was observed. This improvement can probably be attributed to several factors, such as prolonged stimulation of the underlying tissues, enhanced healing processes, or ongoing biological responses initiated by the initial treatments. Given that the control group also showed improvement in the quality of vision over time, there is also a tendency for natural improvement to occur. All these can be reasons for continued improvement in postoperative visual outcomes.

Our study has some limitations. All patients were older Japanese individuals aged 65–88 years. Differences in nationalities and age groups may have influenced the results. The patients were not masked as to which eye was treated with IPL-MGX because a sham mode was not available with the IPL device used (open-label design). Moreover, differences in facilities and medical staff may have influenced the results. We used the Japanese MGD diagnostic criteria, which are designed for all eye doctors, including nonspecialists, to easily diagnose MGD without any special devices [27]. Morphological evaluation using the meiboscore and functional assessments were not implemented because these were not included in the Japanese MGD diagnostic criteria. Future studies should further investigate the relationship between morphological and functional assessments and postoperative visual outcomes. Dry eye was diagnosed based on the Japanese dry eye criteria. These are different from the dry eye criteria of the Dry Eye WorkShop in Western countries [63] because short TBUT-type dry eye occurs more frequently than other types of dry eye in Japan and other Asian countries [64,65]. Therefore, a new diagnostic criterion for dry eye was developed by the Asia Dry Eye Society for Asian populations [66]. The severity of dry eye symptoms does not always accurately correlate with ocular surface deterioration in short TBUT-type dry eye [61]. Therefore, further investigations using other types of dry eye diagnostic criteria are required to determine whether other criteria may influence the results.

A similar observation applies to the C-SPK assessment, which was performed using the basic four-point scale provided by the National Eye Institute/Industry Workshop in the present study. Different evaluation systems for C-SPK, such as the Oxford Scheme [67], may also influence the outcomes. Furthermore, the effectiveness of preoperative MGD-related dry eye treatment with different types of physical treatments, such as IPL alone, MGX alone, and LipiFlow thermal pulsation, has been published [45,46]. However, their effects on postoperative visual outcomes have not yet been reported. Therefore, comparing the effects of different types of physical treatment on postoperative visual outcomes may be valuable. Given the recent wider choice of IOLs, a similar investigation of different types of IOLs is of interest. Finally, preoperative dry eye treatment with LipiFlow thermal pulsation considerably alleviates tear film instability and ocular surface deterioration, not only for MGD-related dry eye but also for dry eye without baseline MGD [45]. Therefore, the effects of preoperative IPL-MGX dry eye treatment on postoperative visual function in dry eyes without baseline MGD warrants further investigation.

## 5. Conclusions

Preoperative MGD-related dry eye treatment using IPL-MGX in cataract surgery with diffractive trifocal IOL implantation is considerably beneficial in alleviating the deterioration of postoperative tear film instability and ocular surface conditions. These factors contribute to the improvement in postoperative visual outcomes such as visual acuity and CS with the use of IPL-MGX. This notable effect of IPL-MGX on CS is more remarkable and lasts longer at higher spatial frequencies, which are affected by both dry eye disease and diffractive multifocal IOLs. Therefore, preoperative IPL-MGX treatment may markedly improve the postoperative quality of vision and increase patient satisfaction, particularly in patients with a combination of dry eyes and diffractive trifocal IOL implantation.

## Figures and Tables

**Figure 1 jcm-13-06973-f001:**
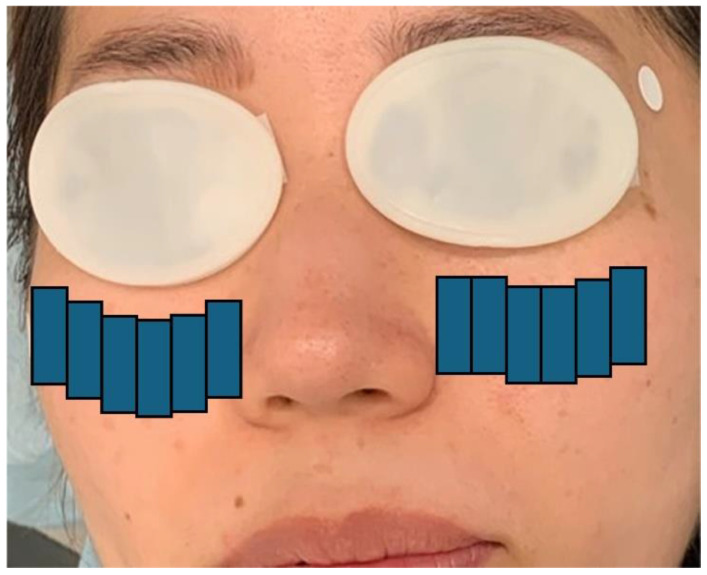
IPL protocol (step 1). Double-pass technique of 12 impacts on the infraorbital/lower eyelid region with a 15 × 35 mm guide light. IPL, intense pulsed light. All patients received treatment for either the right or left eye. This diagram demonstrates the treatment process using eye patches (shown for illustrative purposes only and not depicting an actual patient.

**Figure 2 jcm-13-06973-f002:**
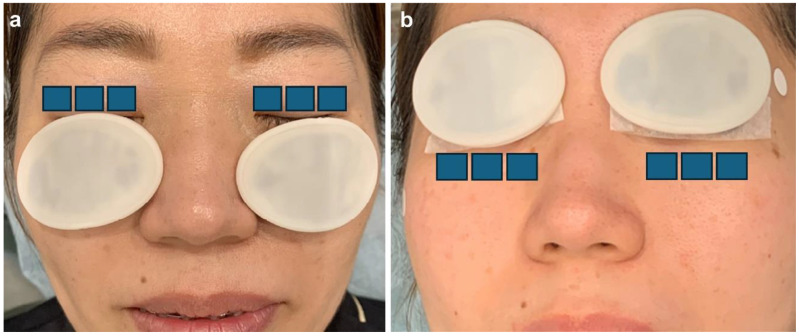
IPL protocol (step 2). Single-pass technique of three impacts on the upper (**a**) and lower (**b**) eyelids with an 8 × 15 mm guide light. IPL, intense pulsed light. All patients received treatment in either the right or left eye. This diagram demonstrates the treatment process using eye patches (shown for illustrative purposes only and not depicting an actual patient).

**Figure 3 jcm-13-06973-f003:**
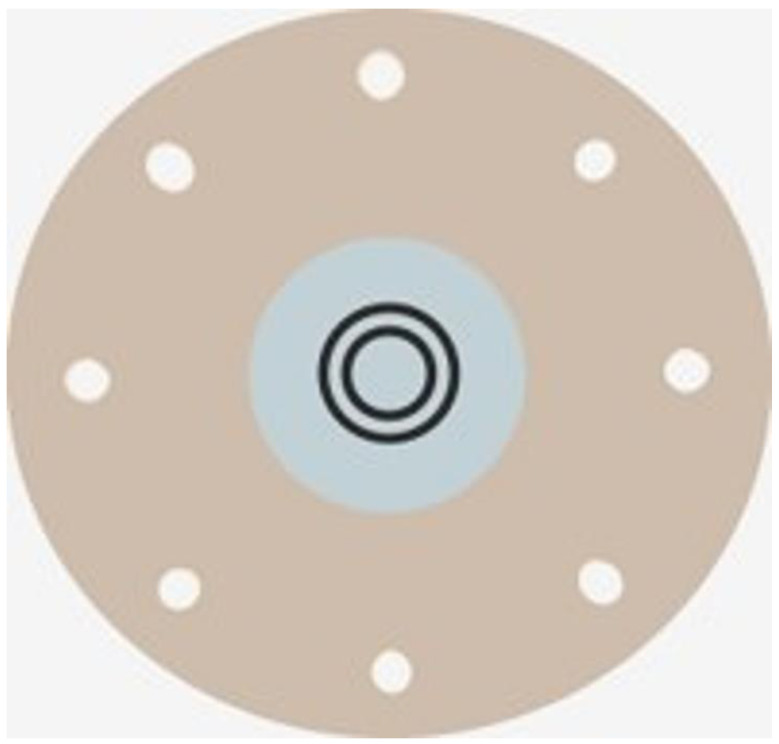
Target rings in the contrast glare tester CGT-1000.

**Figure 4 jcm-13-06973-f004:**
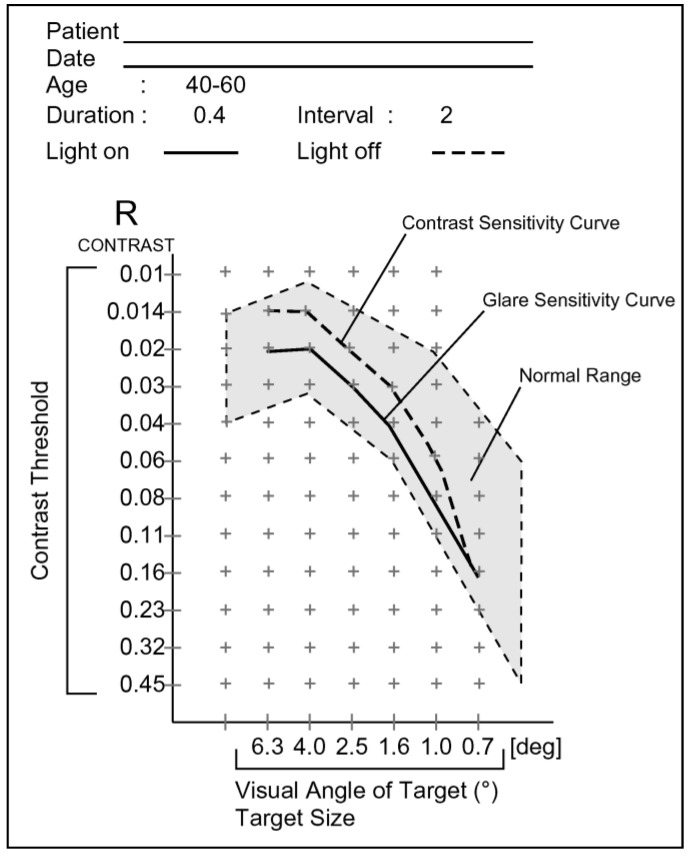
Output data showing the contrast sensitivity against the visual angles of the whole targets.

**Figure 5 jcm-13-06973-f005:**
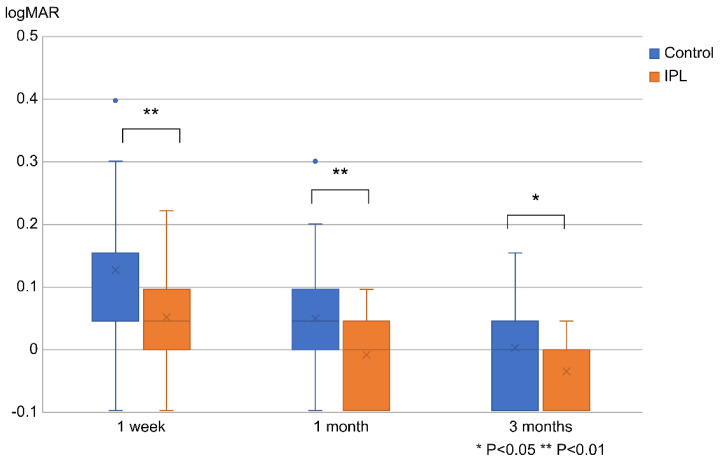
CDVA at postoperative intervals of 1 week, 1 month, and 3 months. The boxplots display the median values and interquartile ranges. CDVA, corrected distance visual acuity; IPL-MGX, combination treatment of intense pulsed light and meibomian gland expression; logMAR, logarithm of the minimum angle of resolution.

**Figure 6 jcm-13-06973-f006:**
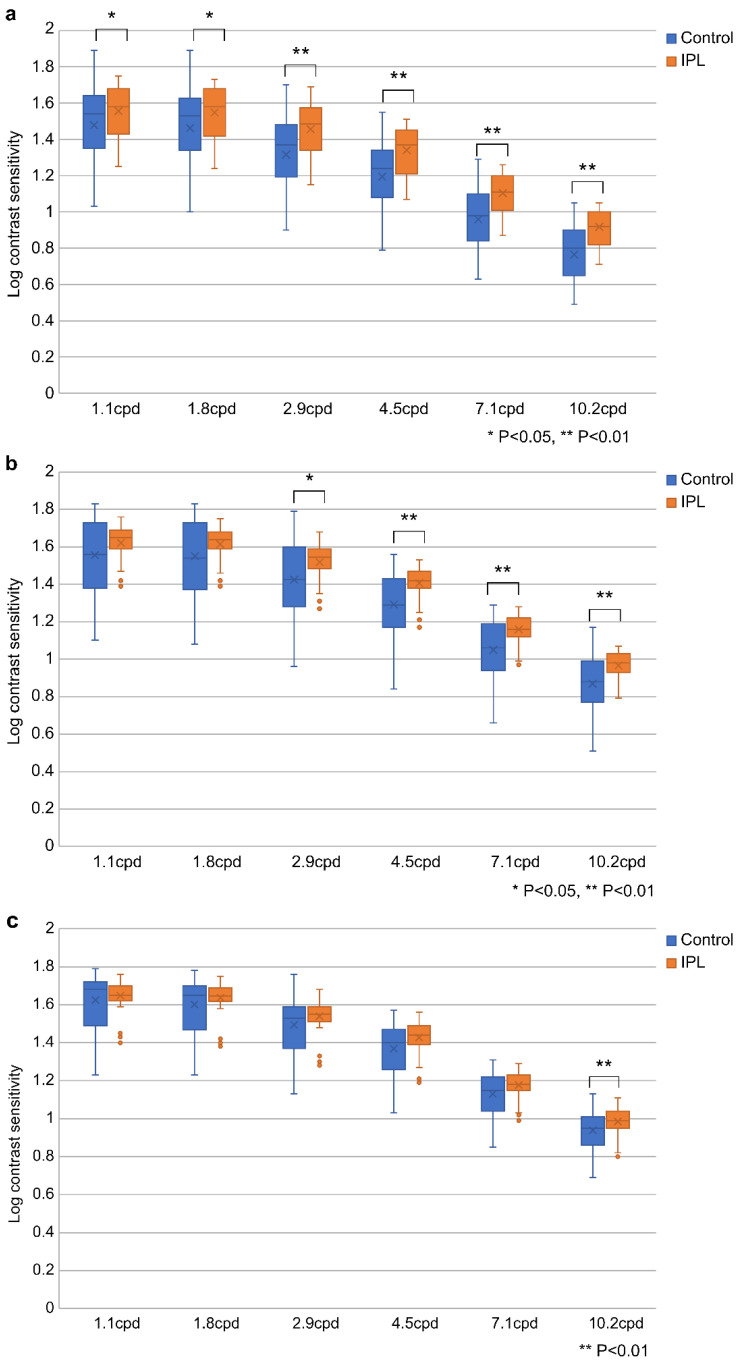
Logarithm of contrast sensitivity at 1 week (**a**), 1 month (**b**), and 3 months (**c**) postoperatively. The boxplots show the median values and interquartile ranges. Cpd, cycle per degree; IPL-MGX, combination treatment of intense pulsed light and meibomian gland expression.

**Table 1 jcm-13-06973-t001:** Baseline data in the IPL-MGX and control groups.

	Control (n = 67)			IPL-MGX (n = 67)			
	Mean	SD	Median	Mean	SD	Median	*p*-Value
Pupil size	7.61	37.50	3.02	3.03	0.15	3.02	1.00
TBUT	3.93	0.82	4.00	3.93	0.86	4.00	1.00
C-SPK	0.51	0.56	0.00	0.52	0.56	0.00	1.00
HOAs	0.27	0.04	0.27	0.28	0.03	0.28	0.65
CDVA (logMAR)	0.44	0.21	0.40	0.45	0.19	0.51	1.00

CDVA, corrected distance visual acuity; C-SPK, superficial punctate keratopathy in the central part of the cornea; HOA, high-order aberration; IPL-MGX, combination treatment of intense pulsed light and meibomian gland expression; logMAR, logarithm of the minimum angle of resolution; SD, standard deviation; TBUT, tear break-up time.

**Table 2 jcm-13-06973-t002:** Mean TBUT, C-SPK, HOA, and CVDA values in the IPL-MGX and control groups after IPL-MGX treatment.

	Control (n = 67)			IPL-MGX (n = 67)			
	Mean	SD	Median	Mean	SD	Median	*p*-Value
TBUT	3.94	0.80	4.00	5.46	0.78	5.00	<0.01
C-SPK	0.52	0.66	0.00	0.30	0.46	0.00	<0.01
HOAs	0.28	0.04	0.27	0.26	0.02	0.25	<0.01
CDVA (logMAR)	0.44	0.19	0.40	0.42	0.17	0.40	1.00

CDVA, corrected distance visual acuity; C-SPK, superficial punctate keratopathy in the central part of the cornea; HOA, high-order aberration; IPL-MGX, combination treatment of intense pulsed light and meibomian gland expression; logMAR, logarithm of the minimum angle of resolution; SD, standard deviation; TBUT, tear break-up time.

**Table 3 jcm-13-06973-t003:** (**a**) Mean values of TBUT, C-SPK, and HOAs before and 1 week after surgery in the control group and (**b**) Mean values of TBUT, C-SPK, and HOAs before and 1 week after surgery in the IPL-MGX group.

(**a**)							
**Control Group**	**Preoperatively**			**1 Week Postoperatively**			
	**Mean**	**SD**	**Median**	**Mean**	**SD**	**Median**	***p*-Value**
TBUT	3.94	0.80	4.00	2.12	0.84	2.00	<0.01
C-SPK	0.52	0.66	0.00	1.54	0.96	1.00	<0.01
HOAs	0.28	0.04	0.27	0.36	0.07	0.34	<0.01
(**b**)							
**IPL-MGX Group**	**Preoperatively**			**1 Week Postoperatively**			
	**Mean**	**SD**	**Median**	**Mean**	**SD**	**Median**	***p*-value**
TBUT	5.46	0.79	5.00	3.96	0.82	4.00	<0.01
C-SPK	0.30	0.46	0.00	1.28	0.79	1.00	<0.01
HOAs	0.26	0.02	0.25	0.33	0.03	0.32	<0.01

C-SPK, superficial punctate keratopathy in the central part of the cornea; HOA, high-order aberration; IPL-MGX, combination treatment of intense pulsed light and meibomian gland expression; SD, standard deviation; TBUT, tear break-up time.

**Table 4 jcm-13-06973-t004:** Mean TBUT in the IPL-MGX and control groups at postoperative intervals of 1 week, 1 month, and 3 months.

	Control (n = 67)			IPL-MGX (n = 67)			
	Mean	SD	Median	Mean	SD	Median	*p*-Value
1 week	2.12	0.84	2.00	3.96	0.82	4.00	<0.01
1 month	2.61	0.87	3.00	4.72	0.79	5.00	<0.01
3 months	3.51	0.88	3.00	5.34	0.66	5.00	<0.01

IPL-MGX, combination treatment of intense pulsed light and meibomian gland expression; SD, standard deviation; TBUT, tear break-up time.

**Table 5 jcm-13-06973-t005:** Mean C-SPK values in the IPL-MGX and control groups at postoperative intervals of 1 week, 1 month, and 3 months.

	Control (n = 67)			IPL-MGX (n = 67)			
	Mean	SD	Median	Mean	SD	Median	*p*-Value
1 week	1.54	0.96	1.00	1.28	0.79	1.00	<0.01
1 month	1.04	0.82	1.00	0.73	0.66	1.00	<0.05
3 months	0.57	0.66	0.00	0.31	0.47	0.00	<0.05

C-SPK, superficial punctate keratopathy in the central part of the cornea; IPL-MGX, combination treatment of intense pulsed light and meibomian gland expression; SD, standard deviation.

**Table 6 jcm-13-06973-t006:** Mean values of HOAs in the IPL-MGX and control groups at postoperative intervals of 1 week, 1 month, and 3 months.

	Control (n = 67)			IPL-MGX (n = 67)			
	Mean	SD	Median	Mean	SD	Median	*p*-Value
1 week	0.36	0.07	0.34	0.33	0.03	0.32	<0.01
1 month	0.32	0.05	0.32	0.29	0.03	0.30	<0.01
3 months	0.29	0.04	0.28	0.26	0.02	0.26	<0.01

HOA, high-order aberration; IPL-MGX, combination treatment of intense pulsed light and meibomian gland expression; SD, standard deviation.

## Data Availability

The raw data supporting the conclusions of this article will be made available by the authors upon request. The data are not publicly available due to ethical restrictions.

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
