# Peer review of "Dry Eye Treatment with Intense Pulsed Light for Improving Visual Outcomes After Cataract Surgery with Diffractive Trifocal Intraocular Lens Implantation"

_jcm, 2024, doi:10.3390/jcm13226973_

Round 1
Reviewer 1 Report
Comments and Suggestions for Authors
-The manuscript is well written, the introduction is appropiate,the results are well tabulated and organised especillay the stress on the contrast sensitivity after cataract with and without dry eye
- The author didn't describe the Types of eye drops used and its frequency.
- The author didn't use Meibobianography?? Which type
- Ethnic differences are present as all patients are japanese and also the age is high ,and lid abnormalities are common in this age
- Why you considered HOA less than 0.5 microns not 0.3 microns as stated for the zerniche analysis.
- Exclusion chreiteria includes previously plugged patients or not
- Any devices to measure qualitative tear film and osmolarity.
- Diabetics should be excluded.
- Why all cases femtolaser and its known that it increase the dryness of the created wounds
- On what bases four sessions for IPL with one week interval?
- Out of 56 reference only six of them are from year 2021
Reviewer 2 Report
Comments and Suggestions for Authors
The authors present a combination of MGD treatment (IPL and MGX) before cataract surgery, in patients with difractive implantation.
The article underlines the importance of alleviating dry eye especially when difractive trifocal implants are used. The results could easily extrapolate to other type of IOLs.
Reviewer 3 Report
Comments and Suggestions for Authors
This study investigates the effects of preoperative treatment for MGD-related dry eye, combining manual meibomian gland expression (MGX) and intense pulsed light therapy (IPL), on visual outcomes following cataract surgery with diffractive trifocal intraocular lens implantation. I have identified some concerns regarding the study design that I believe should be clarified in the manuscript. Please see the enclosed comments, which I hope will enhance the clarity, readability, and overall impact of the manuscript.
Line 24: I suggest adopting a more scientific and indirect writing style, avoiding using personal pronouns such as 'we' or 'our' and opting for an impersonal tone. This recommendation also applies to line 77, line 96, lines 150, 152, and others.
Lines 33-39: I would recommend including p-values for the different analyses presented in the abstract.
Line 53: The acronym "CS" is introduced here but not in the abstract. I recommend ensuring consistent terminology between the abstract and the main text.
Lines 71-73: These sentences can be combined, and the references consolidated, as they convey similar information.
Line 101: Was the same model and type of IOL prescribed for all participants? This is crucial, as CS is a key variable analyzed in the study, and different optical designs or models could have influenced the outcomes, potentially affecting the accuracy of the results.
Line 122: The specific measurement chart or method used to assess visual acuity should be stated.
Line 123: The methods or devices used to measure HOAs should be explicitly mentioned.
Line 138: How was the eye chosen for treatment? Was it randomized, or was the right eye always selected? Please clarify this process in detail.
Lines 137-142: I have ethical concerns about administering a beneficial treatment to only one eye, especially without allowing participants to use eye drops prior to surgery. This aspect should be addressed more thoroughly.
Lines 137-142: The distinction between the control and study groups is unclear. The current explanation is somewhat confusing. Were the 67 participants divided into two groups—one with treated eyes (study group) and the other with untreated eyes (control group)? Or were some participants treated in one eye while the other eye remained untreated? I recommend revising this section for clarity. Additionally, the manuscript states, "A single doctor performed IPL-MGX on the first operated eye but not on the fellow eye undergoing the second operation," while the figures (Figures 1 and 2) show IPL being applied to both eyes, which appears contradictory.
Line 191: The acronym "high-order aberrations (HOAs)" was already defined earlier in the text.
Lines 190-224: There is a notable absence of references in this section, particularly for methodologies and employed techniques, which should be cited to substantiate the study design.
Lines 233-247: Including references for the statistical methods used may be necessary (PMID: 23252852, PMID: 32809534).
Pages 11-13: Please revise the layout as there is significant empty space in this section."
I would strongly recommend that the authors clarify all of these points in order to allow for a proper evaluation of the impact of their results. Other sections, such as the statistical analysis, the results, and even the discussion, appear to have been thoroughly developed and delve deeply into what seems to be the final objective of the work and the current state of knowledge on this topic.
Reviewer 4 Report
Comments and Suggestions for Authors
Authors compare the effect of IPL-MGX treatment on visual outcomes postoperative patients.
So, authors should add MGX in title: Dry eye treatment with meibomian gland expression and intense pulsed light for improving visual outcomes after cataract surgery with diffractive trifocal intraocular lens implantation.
Line 138-140
In the IPL-MGX group patients received 4 treatment sessions.
It means:
1) preoperative: 1 week before the surgery
2) 1week after the surgery
3) 3 weeks after the surgery
4) 5 weeks after the surgery
If these are correct, the patients had not received IPL-MGX treatment for 3 weeks till 3 months examination. However, the visual outcomes more improved at 3 months after the surgery. Please describe the reason for sustained effect of IPL-MGX.
Round 2
Reviewer 3 Report
Comments and Suggestions for Authors
I would like to thank the authors for their efforts in revising the manuscript. The modifications have clarified most of the points previously raised. However, I have a few additional comments that I would appreciate if you could address for further clarification. Additionally, I would like to suggest that changes made in the manuscript be highlighted (e.g., using red text) for future reviews, as indicated in the author's guidelines. This would greatly facilitate the review process.
Line 22: I recommend using the past tense, as the study has already been completed (similar to how the purpose is stated at the end of the introduction). I suggest reviewing this aspect throughout the entire manuscript, particularly in the methodology section.
Lines 156-165: I apologize if I was unclear before. The protocol description indicates that patients receive treatment in one eye, with the other eye serving as a control in a random, alternating manner. However, Figures 1 and 2 seem to show IPL being applied to both eyes. Could you please clarify if these images are meant to be a demonstration (i.e., not depicting an actual patient treated in both eyes)? It is important to ensure this is clear for readers, as this is a key aspect of the study's design.
Lines 235-236: I recommend using the Kolmogorov-Smirnov test in future studies due to the larger sample size; this test is generally more powerful with samples of over 50 participants. However, I suggest reporting the exact p-values of each Shapiro-Wilk test conducted, so that the distribution of each variable is clear and the choice of statistical tests in the results section is well understood.
Lines 239-240: Why were comparisons between sessions not performed, focusing only on comparisons between groups? This is important to understand the study design. Additionally, using the Bonferroni correction for multiple comparisons in the non-parametric analysis could be considered (even if it may not affect the final significance or conclusions of the study, it would enhance the reliability of the results).
Author Response
Detailed response to reviewer comment has been attached herewith.
